# Early Application of ECMO after Sudden Cardiac Arrest to Prevent Further Deterioration: A Review and Case Report

**DOI:** 10.3390/jcm12134249

**Published:** 2023-06-25

**Authors:** Boldizsár Kiss, Bettina Nagy, Ádám Pál-Jakab, Bálint Lakatos, Ádám Soltész, István Osztheimer, Krisztina Heltai, István Ferenc Édes, Endre Németh, Béla Merkely, Endre Zima

**Affiliations:** Heart and Vascular Centre, Semmelweis University, Budapest 1122, Hungary; kiss.boldizsar@med.semmelweis-univ.hu (B.K.); lakatosbalintka@gmail.com (B.L.);

**Keywords:** cardiogenic shock, resuscitation, extracorporeal membrane oxygenation, survival

## Abstract

ECMO has become a therapeutic modality for in- and out-of-hospital scenarios and is also suitable as a bridging therapy until further decisions and interventions can be made. Case report: A 27-year-old male patient with mechanical aortic valve prothesis had a sudden cardiac arrest (SCA). ROSC had been achieved after more than 60 min of CPR and eight DC shocks due to ventricular fibrillation (VF). The National Ambulance Service unit transported the patient to our clinic for further treatment. Due to the trauma and therapeutic INR, a CT scan was performed and ruled out bleeding. Echocardiography described severely decreased left ventricular function. Coronary angiography was negative. Due to the therapeutic refractory circulatory and respiratory failure against intensive care, VA-ECMO implantation was indicated. After four days of ECMO treatment, the patient’s circulation was stabilized without neurological deficit, and the functions of the end organs were normalized. Cardiac MRI showed no exact etiology behind SCA. ICD was implanted due to VF and SCA. The patient was discharged after 19 days of hospitalization. Conclusion: This case report points out that the early application of mechanical circulatory support could be an outcome-determinant therapeutic modality. Post-resuscitation care includes cardiorespiratory stabilization, treatment of reversible causes of malignant arrhythmia, and secondary prevention.

## 1. Introduction

The availability and application numbers of extracorporeal life support (ECLS) and extracorporeal membrane oxygenation (ECMO) are continuously increasing in cardiac surgery centers [1,2]. Even though ECLS and ECMO are complex therapies that require high-level specialized centers, the first multidisciplinary guideline for the organization and application of ECMO therapy for cardiocirculatory support was published in 2021 [3]. The most common indications for ECLS or ECMO are therapy-refractory cardiogenic shock, acute pulmonary embolism, respiratory failure, and hypothermia. ECLS and ECMO have become therapeutic modalities for in- and out-of-hospital scenarios and are also suitable as bridging therapies in emergency situations until further decisions and interventions can be made [1,4].

Based on the international literature, the utilization of ECLS has increased in both in-hospital and out-of-hospital resuscitation in recent years [5,6,7,8]. The Extracorporeal Life Support Organization (ELSO) database reports that the incidence of extracorporeal cardiopulmonary resuscitation (eCPR) increased from 35 cases per year to 400 cases per year between 2003 and 2014 [7]. The current recommendation suggests that eCPR initiation should be considered as a last resort in selected patient populations [5,9].

Based on the length of time that mechanical circulatory support is used, a difference can be made between devices suitable for short-term and long-term support. Short-term devices are used in high-risk percutaneous coronary intervention (CHIP), cardiogenic shock, and post-cardiac arrest, while durable left ventricular assist devices (LVADs) are used in bridge-to-transplant and bridge-to-decision clinical situations, or as a bridge-to-destination therapy. The use of INTERMACS profiles guides the decision of the optimal timing of MCS implantation and the associated risk based on clinical presentation [10].

In addition to ECMO, other new devices for MCS with powerful continuous flow (e.g., Impella^®^) are becoming increasingly available and are in common use in cardiac surgeries and cardiovascular centers. We focus on ECMO in the following [11].

There are two types of peripheral ECMO based on the type of cannulated vessels: venoarterial (VA) and venovenous (VV). Depending on the allocation of the vessels that have been cannulated, VV-ECMO can be further subdivided into femoro-jugular and femoro-femoral VV-ECMO. Both types of peripheral ECMO can provide respiratory support, but the VA-ECMO can assist with acute cardiorespiratory support in cardiogenic shock or cardiac arrest patients as well [12].

Peripheral arterial cannulation is frequently used in non-cardiac surgical scenarios when immediate ECMO support is necessary. The common femoral artery is the most common site for peripheral arterial cannulation in VA-ECMO, but another option can be the axillary artery. Conversely, the aorta serves as the site for the inflow cannula in the case of central VA-ECMO. Implantation of central VA-ECMO is a more invasive procedure that involves opening the chest, although there have been reports of central cannulation using minimally invasive techniques. Surgeons often express concerns regarding the potential risks of bleeding and infection. However, a notable advantage of central VA-ECMO is the ability to achieve sufficient perfusion flow, enabling the use of larger diameter arterial cannulas and the offloading of the left ventricle [13,14].

Peripheral cannulation for VA ECMO is commonly performed by accessing the common femoral artery and vein below the inguinal ligament but above their bifurcations. The size of the arterial cannula typically ranges from 17 to 19 French, which is usually sufficient for providing the required flow based on the patient’s needs. In certain clinical scenarios, such as sepsis, larger cannulas of 19 or 21 French may be necessary to achieve higher flow rates. However, it is important to note that the larger the arterial cannula is, the higher the risk of vascular complication, including limb ischemia. To avoid lower limb ischemia in the case of femoral arterial cannulation, one should add a distally directed shunt, and the perfusion of the distal lower limb should be monitored continuously [13]. While the recommendation of placing the arterial and venous cannulas in separate limbs is based on expert opinion, it is considered preferable in order to minimize vascular complications and facilitate decannulation. Whenever possible, the venous cannula should be placed in the right femoral vein, as it offers the most direct venous path to the inferior vena cava and right atrium [15].

In the context of postcardiotomy scenarios, central cannulation is frequently applied, especially when severe peripheral vascular disease is present at the femoral/iliac levels. Both central and peripheral cannulation approaches have advantages and disadvantages (detailed in Table 1); however, accumulating evidence demonstrates a stronger association between peripheral cannulation and superior overall outcomes [16,17,18].

An important aspect of ECMO treatment is the use of anticoagulant medication. Currently, there are limited high level evidence-based recommendations for the choice of anticoagulant agent (heparin, bivalirudin, argatroban, or nafamostat mesilate) during ECMO treatment. In a recent meta-analysis, bivalirudin was shown to be superior to heparin in reducing mortality and the risk of thrombosis. In the future, anti-adsorbent and anti-coagulant coatings of the MCS and ECLS circuits and bio-hybrid materials could be promising to prevent the initiation of the thrombogenic and inflammatory response [19,20,21,22].
jcm-12-04249-t001_Table 1Table 1Advantages and disadvantages of peripheral and central canulation based on ELSO guideline [23].
AdvantagesDisadvantagesPeripheral cannulationNo surgical incisionLower limb ischemiaLower bleeding riskRetrograde flowNo sternotomyShort-term supportNo cardiac compressionLower flowEasier switch to LVADHigher risk of vascular complicationNo re-sternotomy for removalReduced options for LV ventingCentral cannulationUse of originally implanted cannulasSternotomy to implantAntegrade flowHigher bleeding riskBetter drainageCardiac compressionLong-term supportRe-sternotomy to removeHigher flow Higher infection riskEasier patient mobilizationHigher rate of cerebral embolizationMore options for LV ventingHigher risk of closed aortic valve


The weaning off of VA-ECMO should be contemplated once patients have substantial cardiac improvement, particularly when there is minimal reliance on vasoactive and inotropic drugs to maintain appropriate pulse pressure, mean arterial pressure, and mixed venous/central saturation at a flow rate of 2–2.5 L/min on VA-ECMO. For those patients who fail the weaning process within 5–7 days, temporary LVAD support may be considered to give more time for left ventricular recovery [24].

## 2. Case Presentation

We present a case of a successfully resuscitated 27-year-old male patient due to out-of-hospital cardiac arrest. The patient’s progressive cardiogenic shock and oxygenation failure did not respond to conservative ICU therapy, and he was ultimately stabilized with peripheral VA-ECMO.

The patient suffered sudden cardiac arrest in a public area during his regular 3-hour-long bicycle tour. Approximately after 2 min of no flow anoxic time, a healthcare worker started basic life support (BLS). After 19 min of BLS, the Hungarian National Ambulance Service staff overtook the resuscitation using an automated chest compression device (LUCAS) for a further 39 min of advanced life support (ALS). Overall, after 58 min of cardiopulmonary resuscitation and application of eight direct current shocks, intubation, and mechanical ventilation, spontaneous circulation returned. The Hungarian National Ambulance Service unit transported the patient with continuous intravenous circulatory support to the Heart and Vascular Centre, Semmelweis University, for further treatment (see details in Table 2.).

## 3. Patient Information

In the patient’s medical history, there was a familiar polycystic kidney disease with left-sided dominance and surgical mechanical aortic valve replacement (Sorin^®^, 25 mm, date of implantation: 2017) surgery due to a congenital bicuspid aortic valve. He has been regularly taking acenocoumarin medication only.

## 4. Acute Diagnostic Assessment

As the first step, a cine and coronary angiography were performed: symmetrically opening mechanical aortic valve leaflets without any disabled movement and intact coronary artery anatomy (Figure 1). We ruled out pneumothorax, pericardial, pleural, and abdominal fluid with emergency bedside ultrasound after ICU admission. Echocardiography showed severely reduced left and right ventricular function with diffuse hypokinesis, and mild paravalvular regurgitation of the mechanical aortic valve, and ruled out left ventricular outflow tract obstruction and pericardial fluid. Due to the mechanism of his bicycle fall, prolonged CPR using LUCAS, and therapeutic INR level, a cranial, thoracic and abdominal computed tomography (CT) scan and as well as pulmonary CT angiography (Figure 2) were performed to exclude bleeding and pulmonary contusion and rule out acute pulmonary embolism and aortic dissection. The cranial CT ruled out severe cerebral ischemic damage, postischemic edema, and cerebral herniation.

## 5. Indication of VA-ECMO during Post Resuscitation ICU Care

Due to the rapidly increasing combined catecholamine demand, gas exchange failure with invasive mechanical ventilatory support, and therapy-refractory cardiogenic shock in the first few hours of the post-resuscitation care, the multidisciplinary team decided to immediately implant a peripheral VA-ECMO. Sheaths were placed in the femoral artery and vein in the cardiac intensive care unit to save time, and the peripheral VA-ECMO cannulas were inserted by a cardiac surgeon in the operating room, and mechanical circulatory support was initiated.

Indication was set for short-mechanical circulation support (MCS) as a bridge to a diagnosis and decision due to progressively worsening refractory cardiogenic shock against inotropic and vasopressor support to buy time for ruling out or treating all the potentially reversible or surgically correctable causes [16,24]. Of the commercially available percutaneous temporary MCS, only an intra-aortic balloon pump (IABP) and a VA-ECMO was available in our institute. Even with up-titrated inotropic and vasopressor support, IABP alone is not sufficient for physiologic circulatory maintenance for patients with severe post cardiac arrest biventricular failure due to malignant arrhythmia. VA-ECMO should be considered preemptively in postcardiac arrest shock, particularly to prevent severe multiorgan failure as the consequence of hypoperfusion.

The patient’s young age and practically lack of comorbidities forecast a good prognosis if short term MCS was decided for starting bridging for short decision making or for upgrading to an LVAD or heart transplantation [25].

Aggravating prognostic factors were excluded, such as poor life expectancy, severe liver impairment, acute brain injury, vascular disease, and immuno-deficiency before ECMO application. We used the SAVE score as a decision-support system; the patient scored 7 points (Class I), which predicted 75% in-hospital survival with ECMO (detailed in Appendix A) [26].

## 6. ICU Treatment

During four days of VA-ECMO treatment and five days of invasive mechanical ventilation, combined catecholamine support was successfully withdrawn (Figure 3). Initially, significant elevations in creatine kinase and cardiac enzyme levels were observed, which showed rapidly decreasing kinetics. The high serum lactate levels normalized after ECMO treatment started within hours. Progressive anemia was observed after the start of ECMO, but no transfusion was required, and no source of hemorrhage was confirmed. It was considered to be a side effect of appropriate volume loading for ECMO. The circulation and the target organ functions of the patients were normalized with ECMO support (see details in Figure 4). Intact neurological function was observed during the daily awakenings and after extubation.

Compared to the echocardiogram performed at admission (LVEF: 15%, TAPSE: 12 mm), controls showed improved left and right ventricular function (LVEF: 53%, TAPSE: 22 mm) and proper valvular functions (see details in Figure 5).

After the patient was stabilized, cardiac magnetic resonance imaging (MRI) was performed to clarify the etiology of the cardiac arrest. The cardiac MRI showed mildly reduced left ventricular function, concentric left ventricular hypertrophy, and non-specific patterns of late enhancement in contrast-enhanced cardiac MRI.

An implantable cardioverter defibrillator (ICD) device was implanted due to idiopathic ventricular fibrillation after all reversible causes were excluded.

## 7. Discharge and Follow up

After successful rehabilitation, the patient was discharged to his home after 19 days of hospital treatment. During the ambulatory follow-up visits, no ventricular tachycardia or fibrillation episodes were observed in the device memory (Table 3).

## 8. Conclusions

This case report points out that in the case of a progressively increasing demand for circulatory and respiratory support and therapy-refractory cardiogenic shock, the early application of mechanical circulatory support could be an outcome-determinant therapeutic modality.

The urgently implanted peripheral VA-ECMO was an appropriate clinical, multidisciplinary decision for hemodynamic stabilization, providing extra time to rule out the reversible causes of sudden cardiac arrest and ultimately serving as a bridge to recovery for the patient. Peripheral VA-ECMO does not require re-thoracotomy and has a lower risk of bleeding compared to the centrally inserted ECMO. In clinical situations where the physician is faced with rapidly deteriorating cardiogenic shock despite inotropes and vasopressors, inserting femoral arterial and venous cannulas onsite should be considered in the cardiac ICU or catheterization laboratory to save time.

Post-resuscitation care includes cardiorespiratory care, including the use of ECLS or ECMO if necessary, treatment of reversible causes of malignant arrhythmia, and secondary prevention.

## Figures and Tables

**Figure 1 jcm-12-04249-f001:**
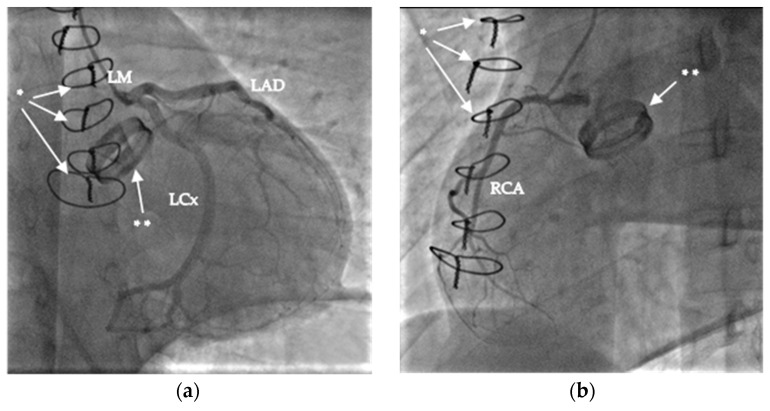
Images of coronary angiography: (**a**) left coronary arteries (left main (LM), left anterior descendent (LAD), and left circumflex (LCx); (**b**) right coronary artery [RCA]. The sternal sutures are marked with *, and the mechanical aortic valve with **.

**Figure 2 jcm-12-04249-f002:**
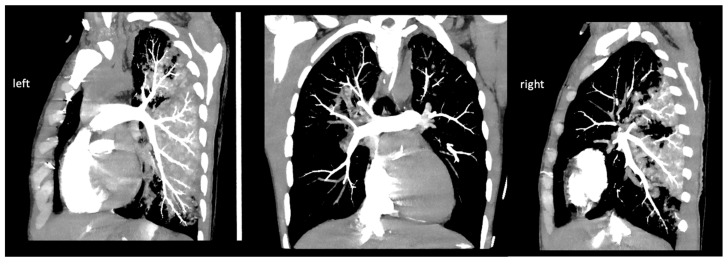
Images (left and right pulmonary artery) of pulmonary CT angiography.

**Figure 3 jcm-12-04249-f003:**
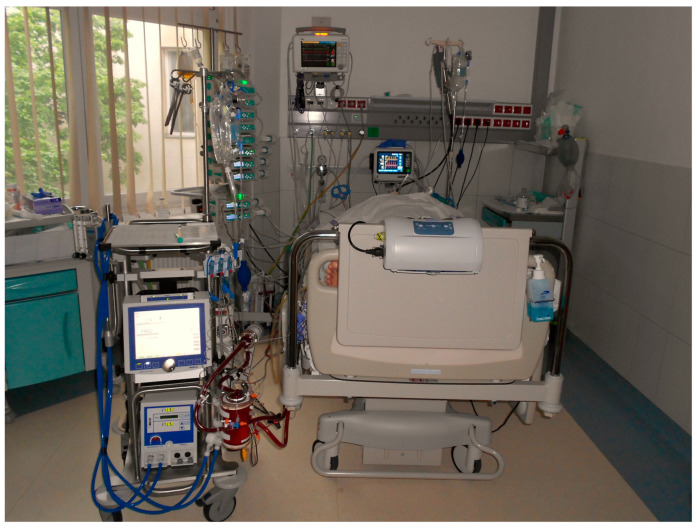
Set up of VA-ECMO treatment in the cardiac intensive care unit.

**Figure 4 jcm-12-04249-f004:**
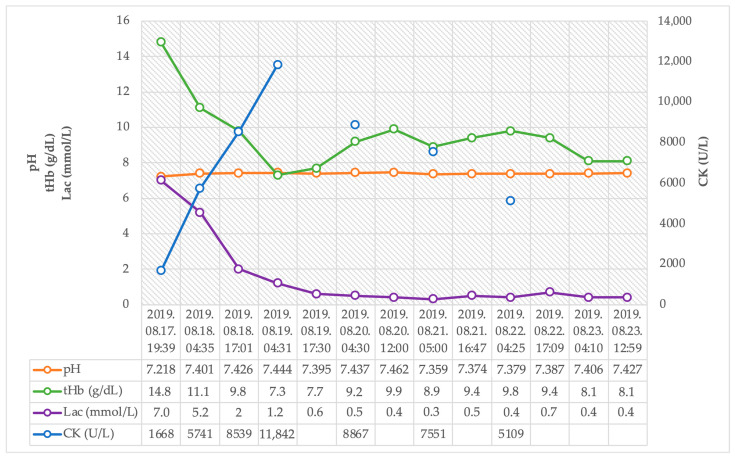
Trends of pH, total hemoglobin, serum lactate, and creatine-kinase after hospital admission.

**Figure 5 jcm-12-04249-f005:**
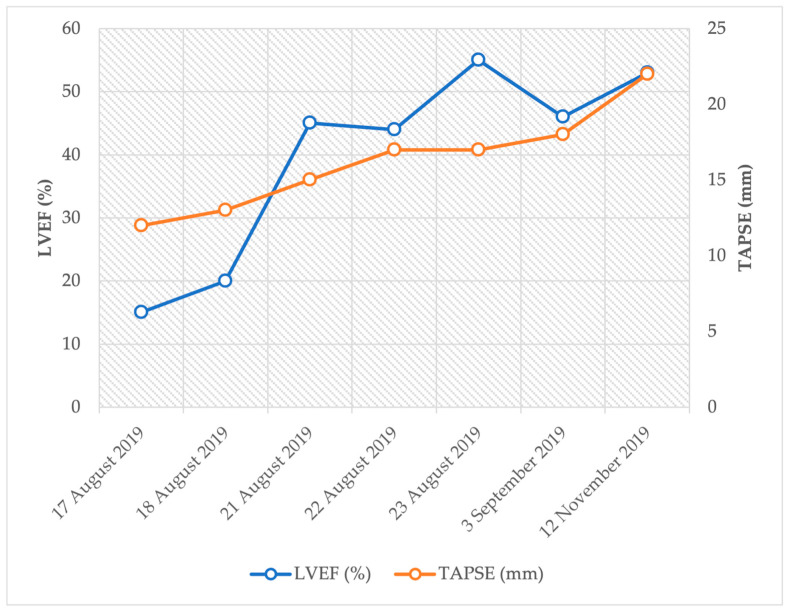
Trends of left and right ventricular function after sudden cardiac arrest.

**Table 2 jcm-12-04249-t002:** Timeline from sudden cardiac arrest to VA-ECMO implantation.

Date	Time	Details
17 August 2019	17:40	sudden cardiac arrest in a public area (OHCA)
	17:42	basic life support started by a healthcare worker
	18:01	advanced life support overtaken by the National Ambulance Service
	18:40	return of spontaneous circulation (ROSC)
	19:16	successful hospital admission to Heart and Vascular Centre Semmelweis University
	19:34	immediate coronary angiogram, negative result
	19:50	bedside emergency heart, lung, and abdominal ultrasound
	20:30	expert transthoracic echocardiography
	21:34	cranial, thoracic, and abdominal CT scan; pulmonary CT angiography to rule out acute aortic dissection and acute pulmonary embolism
18 August 2019	01:30	multidisciplinary decision about the immediate VA-ECMO implantation (progression of cardiogenic shock)
	02:30–03:30	peripheral VA-ECMO implantation

**Table 3 jcm-12-04249-t003:** Timeline of post-resuscitation care, rehabilitation, and follow-up.

Date	Details
21 August 2019	control echocardiography (EF 45%)
17–21 August 2019	weaning from catecholamines
21–22 August 2019	levosimendan infusion (24 h, 0.1 ug/kg/min)
22 August 2019	VA-ECMO explantation after weaning
23 August 2019	extubation, intact neurological function (GCS 15/15)
24 August 2019	active rehabilitation started
4 September 2019	DDD ICD implantation
5 September 2019	discharge to home
12 November 2019	follow-up visit, echocardiography (LVEF: 53%, TAPSE: 22 mm)
20 October 2022	follow-up visit (no arrhythmia episode in ICD memory)

## Data Availability

The data underlying this article will be shared upon reasonable request to the corresponding author.

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
