# Peer review of "Early Application of ECMO after Sudden Cardiac Arrest to Prevent Further Deterioration: A Review and Case Report"

_jcm, 2023, doi:10.3390/jcm12134249_

Round 1

Reviewer 1 Report

1. On line 152, change cardiac necro enzyme levels to cardiac enzymes.

2. In Figure 3, could you add a little space between the baseline (0) and the dates?

3. On line 168, include the values (%) of LVEF and RVEF.

None

Author Response

Thank you for your kind comments and dedicated work with our manuscript! We have revised our manuscript according to your comments and to the best of our knowledge. Please find our answers below to each comment. We hope you will find the answers acceptable.

Point 1: On line 152, change cardiac necro enzyme levels to cardiac enzymes.

Response 1: Thank you for the comment. We corrected the phrase as you recommended. See in line 177.

Point 2: In Figure 3, could you add a little space between the baseline (0) and the dates?

Response 2: Thank you for the comment. I have modified the style of the table so that it might be easier to separate the table and the baseline.

Point 3: On line 168, include the values (%) of LVEF and RVEF.

Response 3: Thank you the comment. We included the values of the LVEF and TAPSE to the main text as you recommended. See in line 190-192.

Reviewer 2 Report

Review for: „Journal of Clinical medicine“ Submission ID jcm-2384365

" Early Application of ECMO After Sudden Cardiac Arrest to Prevent Further Deterioration: A Case Report "

Abstract

Line13

Please explain the abbreviation EMS.

Line 43 - 46

In my view, this paragraph is worded somewhat unclearly. Depending on the vessels that have been cannulated, vv-ECMO can be further subdivided into femoro-jugular and femoro-femoral vv-ECMO. The authors should please include this.

Line 61:

In my view and experience, the use of a 15 F cannula is rarely done. In general, 17-19 F cannulas are used.

Line 63

Here should be a brief reference to the need for a distally directed shunt including the urgent need for continuous perfusion monitoring.

Line 91

ALS is probably meant here. Please correct and explain the abbreviation.

Line 88 - 96

A short note regarding the use of mechanical resuscitation aids (autopulse, LUCAS, etc.) is missing here. This applies in particular to the thoracic trauma suffered with multiple lung contusions.

Line 103

Did they perform a standardized FEEL (Focused Echocardiography in Emergency Life Support) or emergency echocardiography?

If yes what were the findings of the examination?

If no, why did you not perform this examination?

Line 106 - 120

Here, the large pulmonary contusion zones whose cause is traumatic in nature (fall and prolonged CPR) must be mentioned and discussed as a possible (partial) cause of respiratory failure. Especially since prolonged intrapulmonary hemorrhage is to be expected under therapeutic anticoagulation.

can be expected.

Line 122

Why has va-ECMO not been primarily performed in the intensive care unit, especially with regard to the time factor?

Line 129 - 132

Again, the likely traumatic cause, in terms of myocardial contusion during prolonged resuscitation, should be mentioned and briefly discussed.

I.e., the indication for va-ECMO was given on the one hand by the (trauma-associated) cardiogenic shock and the trauma-related lung failure (contusions).

Line 149 - 159

The use of therapeutic hypothermia is controversial, nevertheless the recommendation for mild therapeutic hypothermia is still valid.

To what extent have the authors used such a procedure? Please discuss your approach.

After prolonged resuscitation events, myoglobinemia is found in addition to an increase in CK, as described by you. Have you measured myoglobin? This is highly important, for example, with regard to the development of a CRUSH kidney. Did you require renal replacement therapy (RRT) during the course of treatment and did you apply hemoadsorption with CytoSorb with regard to myoglobinemia?

Did you have any indication of a relevant inflammatory or infectious event during the course of treatment? Did you use antimicrobial chemotherapeutic agents?

I would like to congratulate the authors for your clear and structured presentation of this complex case.

I hope the few comments help to further improve the comprehensibility of this manuscript. In terms of content, I ask you to appropriately describe and discuss the traumatic aspects of the described, from my point of view combined, cardiogenic-pulmonary shock and please discuss the problem of the therapeutic hypothermia.

Only a minor editing of the english language is necessary.

Author Response

Thank you for your kind comments and dedicated work with our manuscript! We have revised our manuscript according to your comments and to the best of our knowledge. Please find our answers below to each comment. We hope you will find the answers acceptable.

Point 1: Please explain the abbreviation EMS.

Response 1: Thank you for the comment. The marked part has been corrected to the term National Ambulance Service, which was used later in the manuscript.

Point 2: Line 43 – 46 - In my view, this paragraph is worded somewhat unclearly. Depending on the vessels that have been cannulated, vv-ECMO can be further subdivided into femoro-jugular and femoro-femoral vv-ECMO. The authors should please include this.

Response 2: Thank you for the comment. We have included the marked part in the manuscript. See line 53-56.

Point 3: Line 61 - In my view and experience, the use of a 15 F cannula is rarely done. In general, 17-19 F cannulas are used.

Response 3: Thank you for the comment, we corrected the this part of the manuscript. See in line 71.

Point 4: Line 63 - Here should be a brief reference to the need for a distally directed shunt including the urgent need for continuous perfusion monitoring.

Response 4: Thank you for the comment. We add a brief scentence based on your comment to the manuscript. See in line 75-77.

Point 5: Line 91 - ALS is probably meant here. Please correct and explain the abbreviation.

Response 5: Thank you for the comment. We corrected and explained the marked abbreviation. See in line 114.

Point 6: Line 88 – 96 - A short note regarding the use of mechanical resuscitation aids (autopulse, LUCAS, etc.) is missing here. This applies in particular to the thoracic trauma suffered with multiple lung contusions.

Response 6: Thank you for the comment. We added a short note about using automated chest compression device. See in line 113-114.

Point 7: Line 103 - Did they perform a standardized FEEL (Focused Echocardiography in Emergency Life Support) or emergency echocardiography? If yes what were the findings of the examination?

If no, why did you not perform this examination?

Response 7: Thank you for the comment. During the resuscitation there was no available ultrasound in the field. After admission we performed emergency heart, lung and abdominal ultrasound. We we have expanded the relevant section in the manuscript. See in line 131-132 and table 2.

Point 8: Line 106 – 120 - Here, the large pulmonary contusion zones whose cause is traumatic in nature (fall and prolonged CPR) must be mentioned and discussed as a possible (partial) cause of respiratory failure. Especially since prolonged intrapulmonary hemorrhage is to be expected under therapeutic anticoagulation.

Response 8: Thank you for the comment. Aggreed with the reiviewer about the pulmonary contusion, but in this case we did not any experienced bleeding from the tracheal tube. We performed chest CT, which ruled out the pulmonary contusion. See in line 138.

Point 9: Line 122 - Why has va-ECMO not been primarily performed in the intensive care unit, especially with regard to the time factor?

Response 9: Thank you for the comment. Maximum sterility can be maintained in the operating theatre according to the local protocol. Operation room and staff was available within 5 minutes altogether with the transportation. Preliminary cannulation was performed in the ICU when the patient's deterioration was initially detected.

Point 10: Line 129 – 132 - Again, the likely traumatic cause, in terms of myocardial contusion during prolonged resuscitation, should be mentioned and briefly discussed. I.e., the indication for va-ECMO was given on the one hand by the (trauma-associated) cardiogenic shock and the trauma-related lung failure (contusions).

Response 10: Thank you for the comment. Free wall rupture was excluded by echocardiography, no pericardial effusion was detected under therapeutic anticoagulation. No imaging modality demonstrating myocardial contusion was performed in the acute phase.

Point 11: Line 149 – 159 - The use of therapeutic hypothermia is controversial, nevertheless the recommendation for mild therapeutic hypothermia is still valid. To what extent have the authors used such a procedure? Please discuss your approach.

Response 11: During ECMO therapy, the body temperature was maintained at 36 degrees Celsius at all times, which corresponds to mild hypothermia.

Point 12: After prolonged resuscitation events, myoglobinemia is found in addition to an increase in CK, as described by you. Have you measured myoglobin? This is highly important, for example, with regard to the development of a CRUSH kidney. Did you require renal replacement therapy (RRT) during the course of treatment and did you apply hemoadsorption with CytoSorb with regard to myoglobinemia?

Response 12: Thank you for the comment. At the time of the case report, myoglobulin level measurment was not available in our central lab. During the ICU therapy the patient did not require renal replacement theapy and we did not apply hemoadsoprtion with CytoSorb.

Point 13: Did you have any indication of a relevant inflammatory or infectious event during the course of treatment? Did you use antimicrobial chemotherapeutic agents?

Response 13: Thank you for the comment. Due to aspiration during resuscitation and infiltrate on pulmonary CT images, empirical antibiotic therapy (piperacillin-tazobactam, vancomycin) was started and de-escalated according to a regional microbiological spectrum (amoxicillin/clavulanic acid). Microbiological cultures were performed, which yielded Staphylococcus aureus in the respiratory tract, which was sensitive to the antibiotic used.

Reviewer 3 Report

I read with great interest the manuscript by Kiss et al. on a case of early application of ECMO after a sudden cardiac arrest of unknown cause.

The case report is interesting and well written. I have only minor issues to assess.

- Line 37-38. The implementation of new mechanical circulatory support devices (doi: 10.1016/j.pcad.2020.09.003 - doi: 10.31083/j.rcm.2018.01.904) and new anticoagulation therapies (doi: 10.3390/membranes11080617 - doi: 10.1111/aor.14276 - doi: 10.1016/j.thromres.2022.02.007) may have also contributed to such incrementation. Please briefly discuss and add these 5 references.

- Line 100. Was 2017 the date of the valve implantation? Please specify.

- In Table 3, the neurological status provided is confusing, as it seems that 3 consecutive assessment provided GCS 4/15, 5/15 and 6/15. Please change into GCS 15/15.

Author Response

Thank you for your kind comments and dedicated work with our manuscript! We have revised our manuscript according to your comments and to the best of our knowledge. Please find our answers below to each comment. We hope you will find the answers acceptable.

Point 1: Line 37-38. The implementation of new mechanical circulatory support devices (doi: 10.1016/j.pcad.2020.09.003 - doi: 10.31083/j.rcm.2018.01.904) and new anticoagulation therapies (doi: 10.3390/membranes11080617 - doi: 10.1111/aor.14276 - doi: 10.1016/j.thromres.2022.02.007) may have also contributed to such incrementation. Please briefly discuss and add these 5 references.

Response 1: Thank you for the comment. We briefly discussed and added the 5 references about the new MCS devices and anticoagulation therapies to the reiveiw part of the manuscript. See in line 43-53, and 87-94.

Point 2: Line 100. Was 2017 the date of the valve implantation? Please specify.

Response 2: Thank you for the comment. Yes, 2017 is the date of the valve implantation. We specified it. See in line 126.

Point 3: In Table 3, the neurological status provided is confusing, as it seems that 3 consecutive assessment provided GCS 4/15, 5/15 and 6/15. Please change into GCS 15/15.

Response 3: Thank you for the comment. We changed the GCS assessment as you recommended in the table 3.